# Effect of Primary Variables on A Confined Plunging Liquid Jet Reactor

**Bader S. Al-Anzi** [1,2]

1   Department of Environment Technologies and Management, Kuwait University, P.O. Box 5969,
    Safat 13060, Kuwait; bader.alanzi@ku.edu.kw; Tel.: +965-9788-5589
2   Department of Mechanical Engineering, Massachusetts Institute of Technology, Cambridge, MA 02139, USA

**Abstract:** The effects of operating conditions including a novel downcomer geometry on the gas/air entrainment rate, $Q_a$, were investigated for a local vertical confined plunging liquid jet reactor (CPLJR) as an alternative aeration process that is of interest to Kuwait and can be used in various applications, such as in wastewater treatment as an aerobic activated sludge process, fermentation, brine dispenser, and gas–liquid reactions. Operating conditions, such as various downcomer diameters ($D_c$ = 45–145 mm), jet lengths ($L_j$ = 200–500 mm), nozzle diameters ($d_n$ = 3.5–15 mm), and contraction angles ($\theta$ = 20–80°), were investigated. A newly designed downcomer with various mesh openings/pores ($D_m$ = 0.25″ (6.35 mm)–1″ (25.4 mm)) was also investigated in the current study. The air entrainment results showed that these were the primary parameters for the measured air entrainment rate in confined systems. The highest gas entrainment rates were achieved when the ratio of the downcomer diameter ($D_c$) to the nozzle diameter ($d_n$) was greater than approximately 5, as long as the liquid superficial velocity was sufficient to carry bubbles downward. Furthermore, a downcomer with mesh openings ($D_m$) less or equal to 0.5″ (12.7 mm) provided a higher entrainment rate than that of conventional downcomer (without a mesh).

**Keywords:** contraction angle; gas/air entrainment rate; confined plunging jet; primary parameters; two-phase; mesh

---

## 1. Introduction

Gas–liquid reactors are employed in a variety of processes, such as aerobic wastewater treatment, air pollution abatement, froth flotation, and fermentation, where the objective is typically to bring two phases into contact to promote mass transfer. The plunging jet reactor concept has been used for several decades to achieve high mass transfer rates by entraining gas bubbles into a liquid at low capital and operating costs [1]. In comparison to conventional sparged systems, such as bubbling gas into a liquid pool, plunging jet devices are able to improve gas absorption rates by creating a fine dispersion of bubbles and increasing the contact time between the gas bubbles and the water at relatively low power inputs. Another new trend of PLJR (plunging liquid jet reactor) is to be utilized as a brine dispenser from desalination plants to mitigate the environmental impact on coastal seawater [2–4].

Plunging jets can be operated either as unconfined or confined devices. In an unconfined system, the liquid jet plunges into an open liquid pool, creating a conical downflow dispersion of fine bubbles that is surrounded by an upflow of larger coalesced bubbles, as shown in Figure 1a [5]. In this case, the bubbles have a small penetration depth because of the spreading of the submerged jet, resulting in a short contact time between the bubbles and the liquid. In a confined system, a downcomer column is used to surround the liquid jet to increase the superficial velocity, as is shown in Figure 1b, and, therefore, the entrained bubbles may be carried to large depths by the liquid downflow. In this study, the liquid superficial velocity is usually sufficiently large to produce a two-phase co-current

downflow, and the majority of entrained bubbles are ejected at the bottom of the downcomer, as is shown schematically in Figure 1b; a fraction of the bubbles will, however, disentrain in the headspace of the confining column, but this gas is available to be re-entrained later. The amount of disentrainment depends on the PLJR primary variables. Thus, a confined plunging liquid jet system is suitable for use with pure gases and may be utilized to improve the mass transfer rate by increasing the jet penetration depth and the contact time between the gas and the liquid.

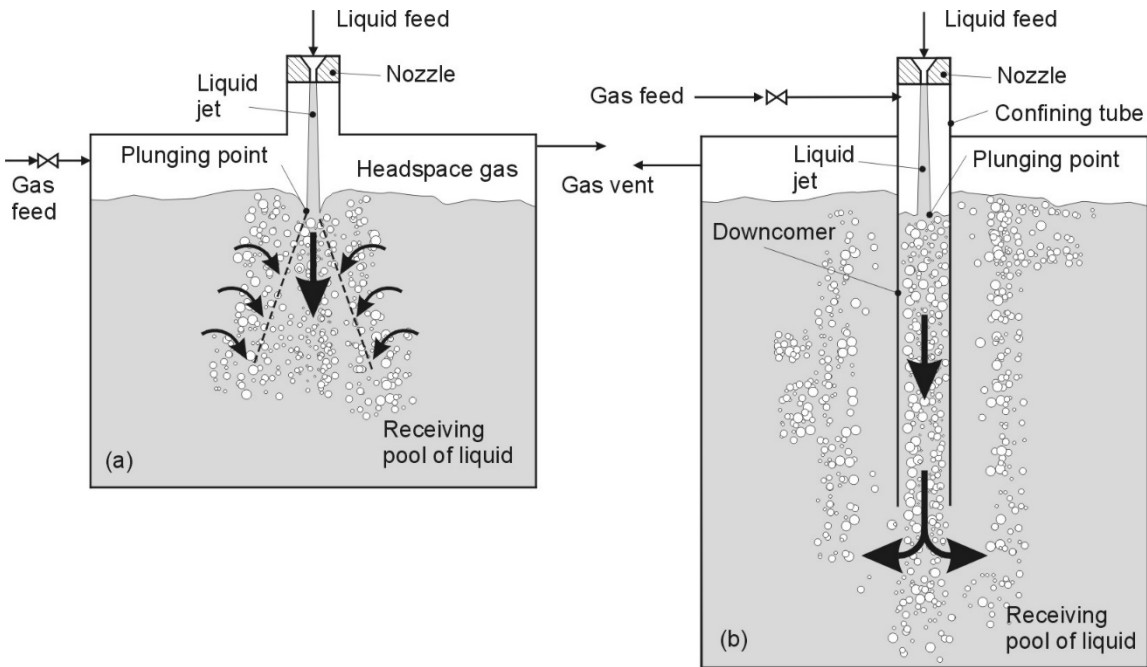

**Figure 1.** (**a**) An unconfined plunging jet system; (**b**) the same jet surrounded by a confining downcomer column [5].

Unconfined plunging jet systems have been extensively studied by many authors: Bin [6] provided a comprehensive review that covers much of the earlier work in this area. Above a minimum entrainment velocity, the plunging jet deforms the surface of the receiving pool, and gas is trapped as the jet passes through the liquid surface. For vertical water jets, the minimum entrainment velocities, also known as onset velocities, reported are in the range of 1.5–2.5 m/s [7–10]. For example, Van de Donk [11,12] investigated the effect of the primary variables—the jet length, $L_j$, the roughness of the jet surface (which depends on the nozzle design and the jet Reynolds number, $Re_j$), the jet velocity, $V_j$, and the nozzle diameter, $d_n$—on the volumetric gas entrainment rate, $Q_a$. The author found that for relatively low-velocity turbulent jets, the entrainment rate was irregular and that surface disturbances controlled the entrainment process. For relatively high jet velocities, air was entrained within the jet before the plunging point, forming a boundary layer that carried the gas below the water surface. Therefore, at low jet velocities, entrainment primarily occurs because of jet envelopment and roughness, whereas at higher jet velocities, entrainment occurs because of jet surface roughness and by air being drawn down through the boundary layer. Roy et al. [13] utilized a flow visualization system in an unconfined liquid jet to report the formation of air sheath, two-phase flow motion, and the random motion of a single bubble. Bin [6] developed the following correlation for the volumetric gas entrainment rate for a vertical jet plunging into an unconfined receiving pool based on a fit to Ohkawa et al. [14] data:

$$\frac{Q_a}{Q_l} = 0.016 \left[ Fr_j^{0.28} \left\{ \frac{L_j}{d_n} \right\}^{0.4} \right]^{1.17},$$

(1)

In Equation (1), the Froude number is given by $Fr_j = \frac{V_j^2}{gd_n}$, where $V_j$ is the jet velocity and the liquid volumetric flow rate is given by $Q_l = V_j \pi d_n^2/4$. Equation (1) shows that the entrainment rate increases with the jet velocity, the jet length, and the nozzle diameter for an unconfined system.

Another group of authors, such as Harby et al. [15], investigated the effects of different ranges of operating variables on the flow characteristics and their flow patterns through conducting a series of experimental work for short nozzles of small length-to-diameter ratios ($L_j/d_n \leq 5$). They found that the penetration depth of the entrained bubbles depends largely on the downward velocity field in the two-phase region.

Kramer et al. [16] performed experimental investigations and mathematical modeling of the penetration depth of plunging water jets. Their results showed an increase in penetration depth with increasing momentum flows and decreasing jet lengths, which agrees with previous findings in the literature.

Yamagiwa et al. [17] studied the entrainment of gas into a downflow bubble column, which was effectively a confining downcomer. The authors showed that the gas entrainment rate increased with the jet velocity, the jet length, and the nozzle diameter. The authors found, however, that the gas entrainment rate was not strongly affected by either the column diameter or the downcomer length, and, therefore, did not include these variables in the correlation developed, which is given in Equation (2) below:

$$\frac{Q_a}{Q_l} = 2.24 \times 10^{-3} \left(Fr_j\right)^{0.4} \left(Re_j\right)^{0.26} \left(\frac{L_j}{d_n}\right)^{0.48},\tag{2}$$

where the Reynolds number is given by $Re_j = \frac{(\rho_l V_j d_n)}{\mu_l}$, where $\mu_l$ is liquid viscosity Ohkawa et al. [10] studied the effects of a confining column with long downcomer lengths, $H_c$, on $Q_a$, and developed the following correlation:

$$Q_a = 0.968 \left(V_j^3 d_n^2\right)^{0.8} \left(\frac{d_n}{D_c}\right)^{1.3} \left(\frac{D_c}{H_c}\right),\tag{3}$$

In Equation (3), the downcomer diameter has a fairly weak effect, and the downcomer length has a much stronger effect on $Q_a$ than in Yamagiwa et al. [17] correlation. For a given set of operating conditions and vertical jets, Yamagiwa's correlation (Equation (2)) predicts a higher entrainment ratio, $Q_a/Q_l$, than that predicted by correlations for unconfined jets, e.g., Equation (1). Ohkawa et al. [14] reported that systems with a downcomer produced gas entrainment rates that were 25%–60% larger than those produced by unconfined jets operating at comparable conditions. However, it is difficult to compare results obtained by different work with each other directly, often because the nozzle internal design affects the jet turbulence intensity and, therefore, the surface roughness, thereby significantly affecting the gas entrainment rates. However, there is evidence in the literature that the presence of a confining column not only increases the penetration depth of the column but may also affect the flow rate of the entrained gas.

The vast majority of the work on plunging jet entrainment mechanisms has been conducted on unconfined plunging liquid jets, whereas relatively little has been reported on confined plunging jet reactors. Low [18] noted that confined and unconfined systems share rather similar entrainment mechanisms, the main difference being that in the former system, the flows in the receiving pool are restricted by the proximity of the downcomer wall. Ajay Mandal [19] used a confined system to generate fine down-flow bubbles that resulted in significantly higher gas-hold up than other gas–liquid systems. A number of other authors have studied the bubble penetration depth behavior of neutrally buoyant plunging jets in water, where the jet and the receiving pool were of the same density [9,20–22].

The current study aims to investigate the effect of a wide range of primary variables, some of which, such as $V_j$, $L_j$, $d_n$, and $D_c$, have been investigated previously by Al-Anzi et al. [5,20]. However, in this study, different dimensions of the primary variables were used along with the new apparatus configuration that was manufactured locally, as described in the Materials and Methods section. The purpose of this is to make sure that the system is running efficiently before carrying out further

experiments and confirm previous studies. Then the effect of two new variables, contraction angle ($\theta$) and mesh/sieve opening, on the net air entrainment flow rate were carried out to develop a deeper understanding of the mechanisms and phenomena that occur within the confining tube as the jet velocity increased. The results of this study will enable a designer to choose the best combinations of operating conditions that provide a high air entrainment rate at minimum cost and can be used in future applications to enhance the oxygen mass transfer rate at a reasonable power input.

## 2. Materials and Methods

Figure 2 shows a confined plunging liquid jet apparatus that has been constructed at the Kuwait University (College of Life Sciences). A centrifugal pump was used to draw tap water from the base of a $1.5 \times 0.5 \times 0.5$ m³ reservoir and feed it to a nozzle to form a liquid jet. The maximum liquid flow rate that could be pumped was 25 L/min; a rotameter was used to measure the liquid flow rate. The internal nozzle geometry was similar to that used by Ohkawa et al. [14] and recently by Al-Anzi et al. [5] that consisted of a straight cylindrical section with a length to diameter ratio of 5 (Figure 3).

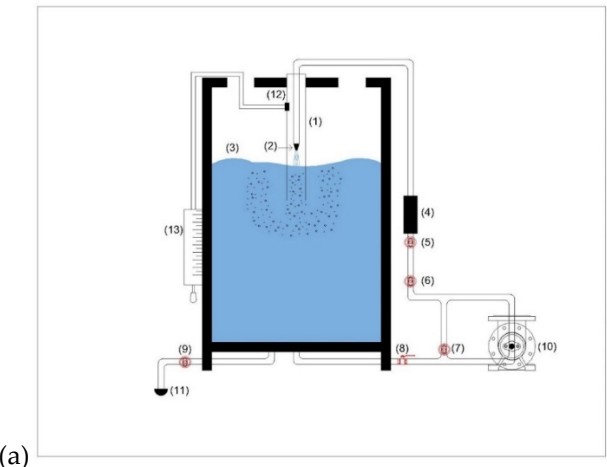

(a)

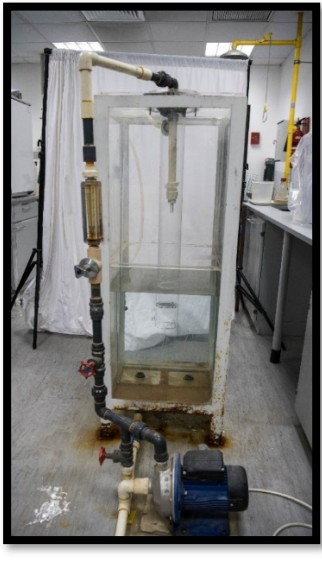

(b)

**Figure 2.** (**a**) Schematic of the confined plunging liquid jet experimental apparatus with the parts marked as follows: (1) Downcomer, (2) Nozzle (3) Water tank, (4) Rotameter, (5–9) Valves, (10) Pump, (11) Outlet, (12) Air tapping, and (13) Bubble meter. (**b**) Plunging Liquid Jet reactor built in the laboratory, as designed.

However, the inlet section converged at 20°, 30°, 50°, 60°, and 80° angles ($\theta$) as indicated by Figure 3, which also shows the nozzle geometry. Nozzle exit diameters of $d_n$ = 3.5, 6, 8, 10, and 15 mm were used in the current set of experiments. The nozzle exit was placed concentrically inside the downcomer column and at various distances from the surface of the receiving pool to produce jet lengths of $L_j$ = 200, 300, 400, and 500 mm. In all of the cases studied, the jet length was less than the break-up length, such that entrainment occurred by a continuous stream of liquid impinging on the receiving pool surface.

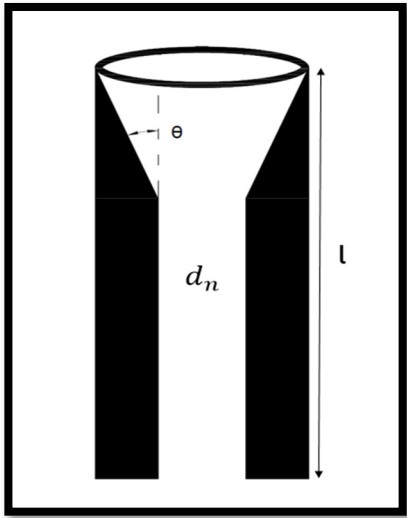

**Figure 3.** Nozzle angle ($\theta$), inner diameter ($d_n$), and nozzle length (l).

Various downcomers with diameters of $D_c$ = 45, 65, 95, and 145 mm and a total length of 600 mm, and Mesh openings ($D_m$) = 0.25″ (6.35 mm), 0.5″ (12.7 mm), and 1″ (25.4 mm) (Figure 4) were used in the current study. The mesh was fixed at a distance of 10 cm below the receiving pool surface, as shown in Figure 5. The volumetric air entrainment rate, $Q_a$, was measured using a soap bubble meter, which provided negligible resistance to the flow. The soap bubble meter comprised a cylindrical tube with an inner diameter of 73 mm and a length of 1000 mm. The soap Bubbles were generated inside the bubble meter using a solution made of 10% household detergent and 5% glycerin, and the rest was water.

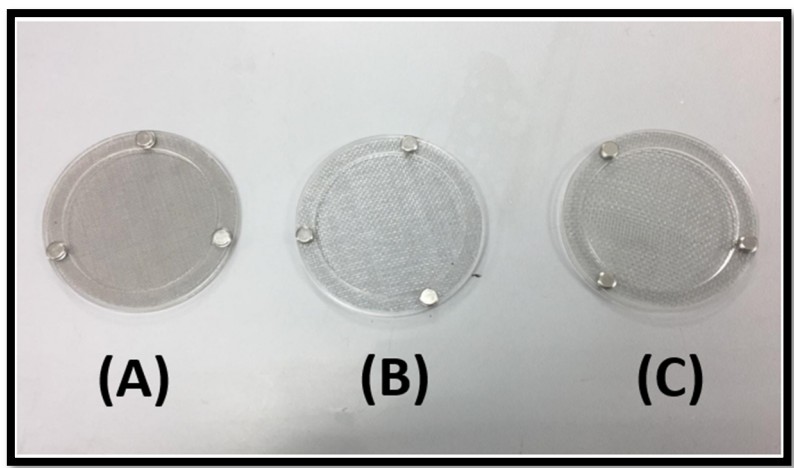

**Figure 4.** Mesh openings of different dimensions (A) $D_m$ = 0.25″ (6.35 mm), (B) $D_m$ = 0.5″ (12.7 mm), and (C) $D_m$ = 1″ (25.4 mm) used in the study.

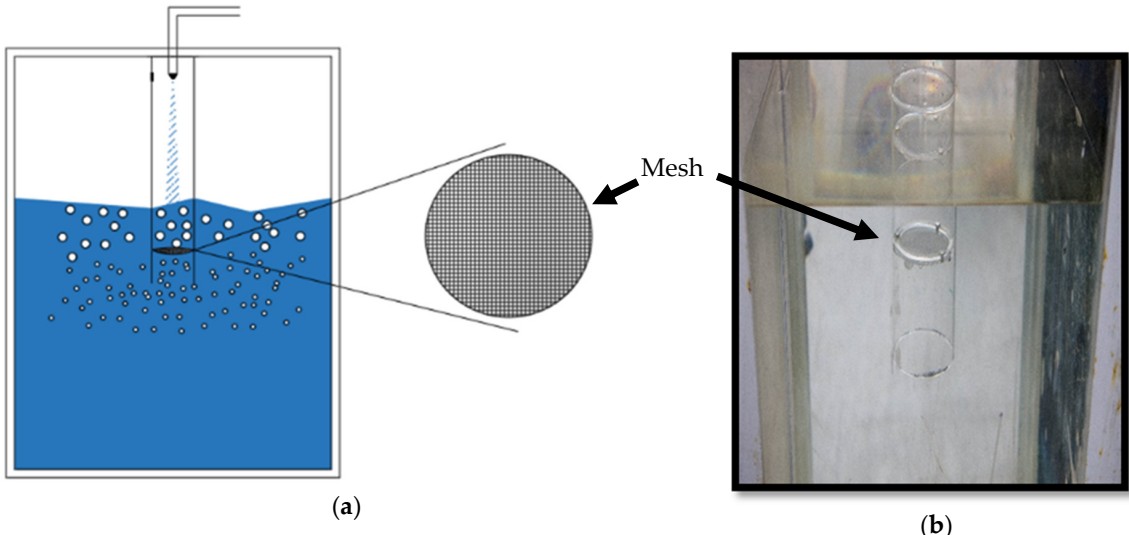

**Figure 5.** (**a**) Schematic diagram of the mesh in the downcomer (**b**) Position of mesh in the downcomer, Mesh depth between mesh and receiving pool is 10 cm.

In this study, the air entrainment rate was measured as the net flow of gas removed by the jet from the headspace. Note that some air can be entrained and then disentrained back into the headspace, and only the net flow rate is measured by the soap bubble meter, i.e., the air flow rate leaving the bottom of the downcomer. This effect becomes more significant as the downcomer diameter, the jet velocity, and the downcomer immersion depth increase, resulting in a greater degree of disentrainment. In contrast, the gas flow rate is generally only a measure of the gross entrainment rate, e.g., Equation (1), in unconfined systems.

## 3. Results and Discussion

### 3.1. Observations

The jet was smooth at low jet velocities, but as $V_j$ increased, the flow became more turbulent, and the surface roughness increased. Low-velocity jets caused a depression of the free surface at the plunging points, entraining bubbles from the jet periphery via mechanisms similar to those previously reported by McKeogh and Ervine [21] and Bin [22] for unconfined systems. However, the radial inflow to the jet periphery was affected by the proximity of the confining walls. In many of the experiments reported here, a vortex formed around the plunging point, which affected the shape of the surface depression around the jet. At higher $V_j$ values, the surface of the jet became rough, and pockets of gas were occluded as the liquid jet passed into the receiving pool. The large entrained gas bubbles were subsequently broken up by the shear flows surrounding the plunging point. Evans and Jameson [23] described distinct regions of the flow that were also experimentally observed in this study. The primary features of the flow were a mixing region in the jet center-line in which the two-phase jet rapidly expanded until the jet hits the walls surrounding this region.

### 3.2. Effect of Nozzle Contraction Angle on Air Entrainment Rate

Most of the researchers working on the PLJR have overlooked the effect of the nozzle contraction angle ($\theta$) on the air entrainment rate $Q_a$. Al-Anzi [20] theoretically studied the effect of the nozzle contraction angle and showed that increasing $\theta$ by 45° resulted in an insignificant decrease in the pressure drop ratio. This result was confirmed by Yamagiwa et al. [24] experimental observations that the power efficiency was independent of $\theta$, where the increase in the air entrainment rate with $\theta$ was attributed to the surface roughness of the jet. However, these results are in disagreement with other results reported in the same study on the effect of $\theta$ on the jet surface roughness for a given

dimensionless nozzle length (nozzle length/nozzle diameter), which showed that the dimensionless jet surface roughness was constant for $\sin \theta > 0.6$ ($\theta > 50°$), i.e., the air entrainment rate was constant.

To further investigate this phenomenon, Figure 6 shows the effect of the nozzle contraction angle on the air entrainment rate carried out in this study for $\sin \theta > 0.6$ by varying the jet velocity, $V_j$, for each dataset and maintaining the remaining primary variables ($d_n$, $D_c$, $L_j$, and $l_n/d_n$) constant. Generally, the dimensionless air entrainment rate ratio $Q_a/Q_j$ increased with the nozzle contraction angle because of the increase in the surface roughness. The air entrainment rate ratio, $Q_a/Q_j$, increased linearly with the nozzle contraction angle at high jet velocities; however, the linearity of the fit became poorer as the jet velocity decreased, as indicated by lower values of the correlation coefficient, $r^2$. The slope of the line for each dataset was not constant and increased with the jet velocity. This result differs from that of Yamagiwa et al. [24] where the slope of the line was forced to be constant by manipulating the operating variables for each individual dataset.

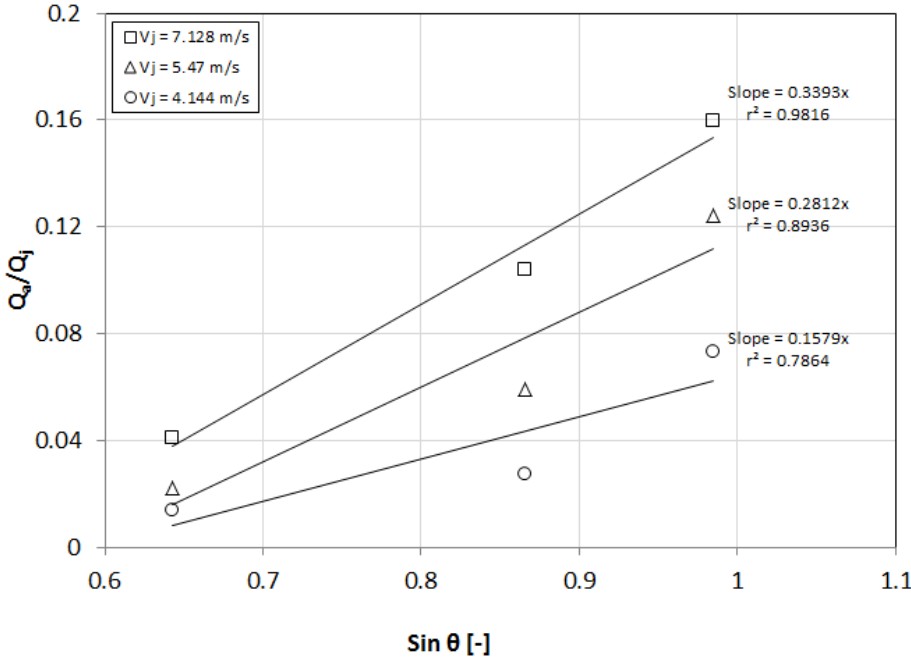

**Figure 6.** Linear relationship between dimensionless air entrainment rate and nozzle contraction angle.

An exponential curve fits the current data more accurately than a linear fit, as shown in Figure 7, where the correlation coefficient $r^2 > 0.92$. Therefore, the air entrainment rate can be described using the following empirical equation, where

$$A = (0.7 - 3.1 ) \times 10^{-3} \text{ and } B = 4.02 - 4.94, \tag{4}$$

For $\sin \theta < 0.6$ ($\theta < 50°$), the air entrainment rate increased, corresponding to inconsistent behavior that may have been caused by human or systematic error; therefore, the data should be re-evaluated by carrying out more experiments.

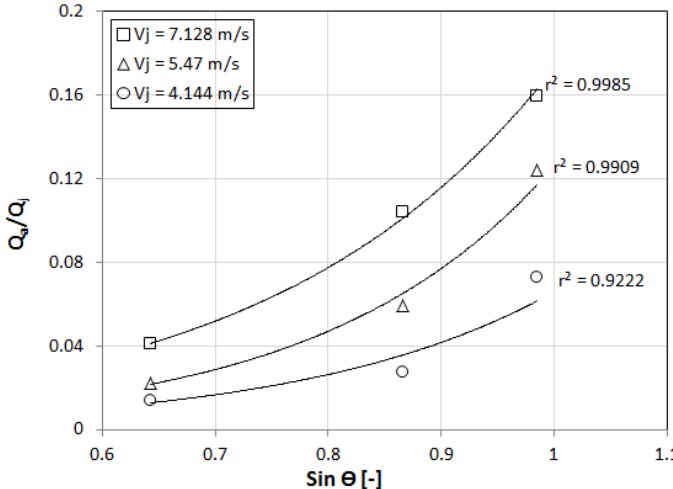

**Figure 7.** Exponential fit between dimensionless air entrainment rate and nozzle contraction angle.

### 3.3. Effect of Nozzle Diameter on Gas Entrainment Rate

Figure 8 shows the effect of the nozzle diameter on the air entrainment ratio for a 45 mm diameter downcomer and a jet length of 500 mm; similar results were obtained for other combinations of jet lengths and downcomer diameters. Figure 8 shows that for a given jet velocity and the same operating conditions (i.e., the same jet length and downcomer diameter), the entrainment rate increased significantly with the nozzle diameter. This result agrees with the findings of Al-Anzi et al. [5]. However, the liquid volumetric flow rate produced by the pump was limited; therefore, the highest jet velocities (and consequently some of the highest values of $Q_a$) could only be obtained using the smaller nozzle diameters.

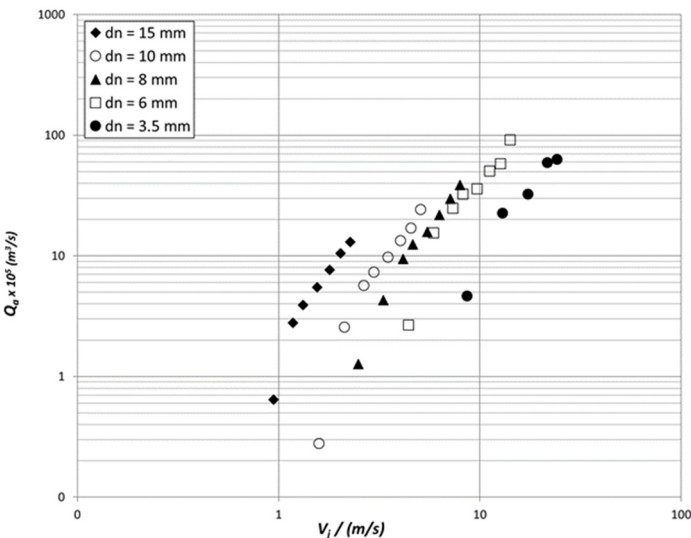

**Figure 8.** Effect of nozzle diameter on gas entrainment rate for a 600-mm downcomer length, $L_j = 500$ mm, and $D_c = 45$ mm.

All the correlations in the literature, i.e., Equations (1–3), show that the entrained gas rate should increase with the nozzle diameter. For a given jet velocity, increasing the nozzle and, therefore, the jet diameter produces a larger jet perimeter at the plunging point, thereby entraining more gas. On this basis, the entrainment rate is expected to be approximately proportional to the nozzle diameter; however, the correlations Equations (1) and (2) suggest a slightly larger exponent of 1.2 to 1.4.

### 3.4. Effect of Downcomer Diameter on Gas Entrainment Rate

Figure 9a,b, respectively, shows two representative datasets for the 500-mm jet length and nozzle diameters of 6 and 8 mm, corresponding to the same downcomer. Both sets of results show that the air entrainment rate increased with the downcomer diameter until it reached a maximum and then either decreased or leveled off, depending on the jet velocity. At low jet velocities and for larger downcomer diameters, the superficial liquid velocity in the downcomer was insufficient to carry all of the entrained bubbles downwards. Thus the *net* entrainment rate decreased: A significant fraction of the bubbles was entrained and then rose in the recirculation eddy and disentrained at the free surface. In the regions where the gas entrainment rate decreased as the downcomer diameter increased, the liquid superficial velocity was less than approximately 0.20–0.25 m/s, which corresponded to a typical bubble terminal velocity. Thus, a fraction of the entrained bubbles was able to rise against the down flowing liquid and be disentrained.

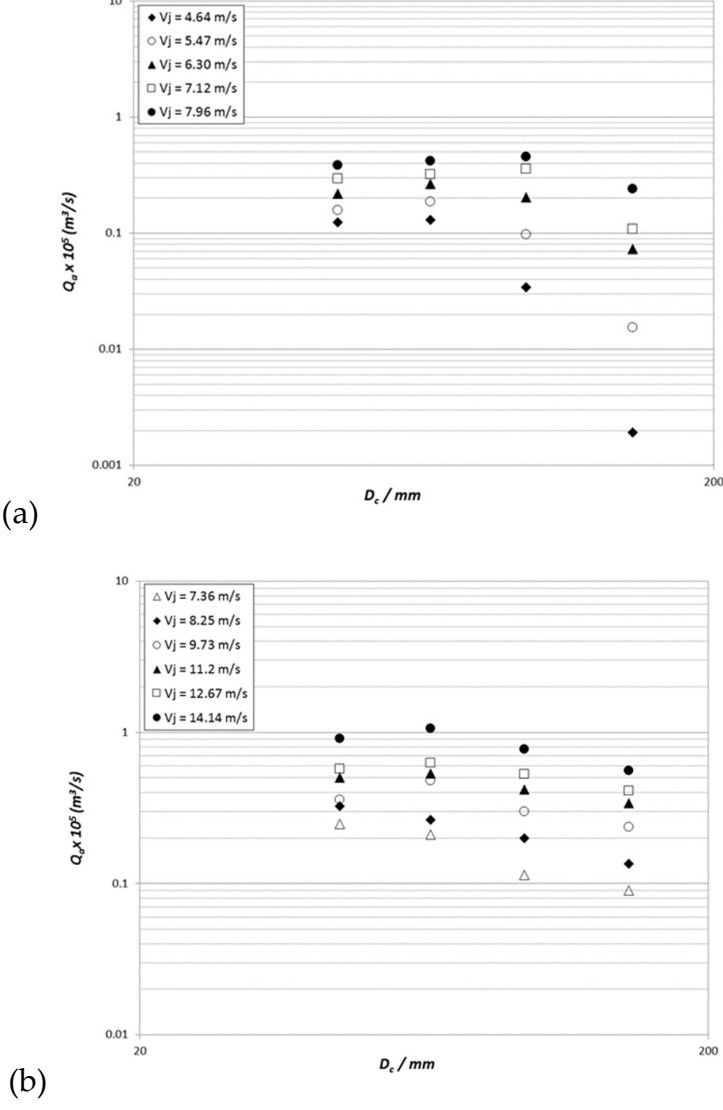

**Figure 9.** Effect of downcomer diameter and jet velocity on the entrained gas flow rate for a 600-mm downcomer length, $L_j$ = 500 mm, and (a) $d_n$ = 8 mm, (b) $d_n$ = 6 mm.

At the higher jet velocities, the maximum value was measured when the ratio of the downcomer diameter ($D_c$) to nozzle diameter ($d_n$) was approximately 5.

### 3.5. Effect of Jet Length on Gas Entrainment Rate

As described above, increasing the jet length should increase the amplitude of the disturbances on the surface of a rough jet, because these disturbances have a longer time to grow since their inception at the nozzle exit. Thus, increasing the jet length should increase the gas entrainment rate in the confined plunging case. Figure 10 shows this trend, where the entrained gas flow rate increased by approximately 56% when the jet length doubled from 200 to 400 mm, while the jet velocity was maintained approximately constant. Similar results were obtained for other nozzle and column diameters; this effect was significant but not large, probably because of the relatively short jet lengths used in this portion of the study. By comparison, Equations (1) and (2) predict an increase in the entrainment rate of approximately 40% for a two-fold increase in the jet length, whereas Equation (3) predicts no effect, which simply reflects the fact that this variable did not appear to be controlled or measured during Ohkawa et al. [14] experiments; in other studies, the same research group showed that $Q_a$ depended on the jet length.

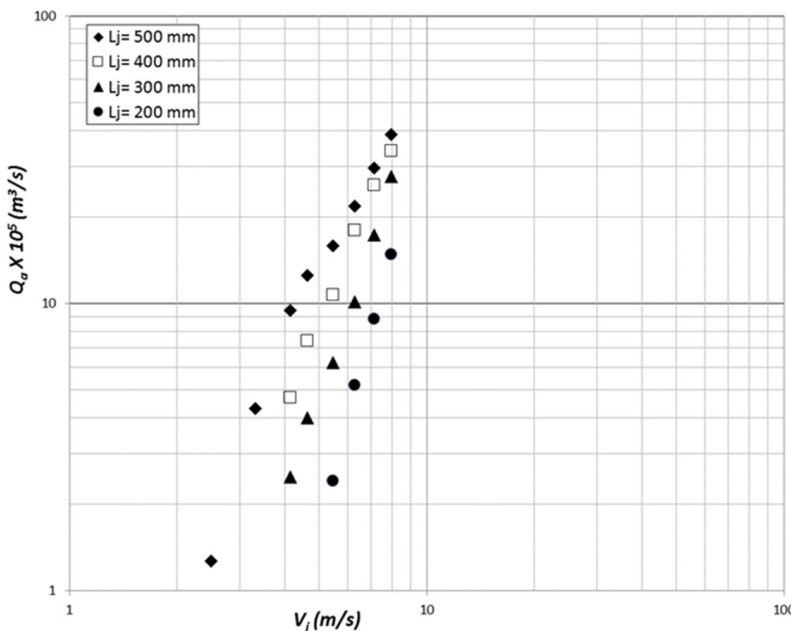

**Figure 10.** Effect of jet length ($L_j$) and jet velocity on entrained gas flow rate for a 600-mm downcomer length, $d_n = 8$ mm and $D_c = 45$ mm.

### 3.6. Effect of Mesh/Sieve Pore Size on Net Air Entrainment Rate

Figure 5 shows a novel downcomer that incorporates a mesh at a desired distance from the receiving pool surface. The purpose of this work is to increase the bubble penetration depth by breaking the bubble swarm into a uniform bubbly downflow comprising fine bubbles. This, in turn, delayed the coalescence of fine bubbles into larger ones and, hence, increased the contact time and reduced the disentrainment rate that increased the measured net entrainment rate (bubbles leaving at the bottom of the downcomer).

Meshes with opening diameters of $D_m = 0.25''$ (6.35 mm), 0.5″ (12.7 mm), and 1″ (25.4 mm) were used in the current study to investigate mesh effects on gas entrainment rate when compared with conventional downcomer (without mesh). Results showed that the gas entrainment rate increased with mesh openings ($D_m$) at high jet velocities (Figure 11 and Figure S1).

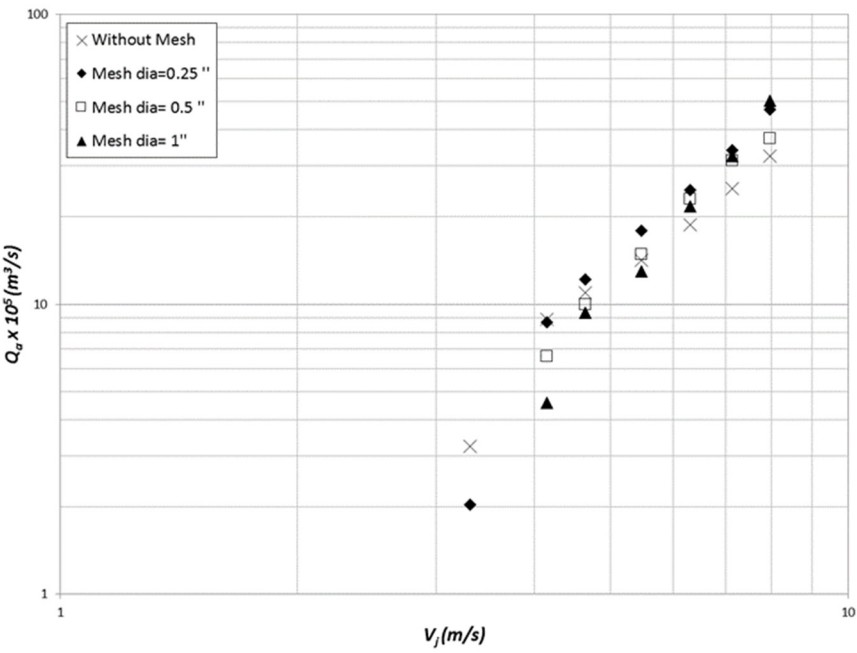

**Figure 11.** Mesh effect on jet velocity on entrained gas flow rate.

## 4. Conclusions

Several measurements were made to investigate the effects of the nozzle contraction angle, jet velocity, the jet length, the nozzle diameter, and the downcomer diameter on the volumetric rate of gas entrainment in a confined plunging jet system. The use of a confining column clearly significantly increased the bubble penetration depths attained. The entrainment rate increased exponentially with nozzle contraction angle greater than 50°. The entrainment rate increased with the jet velocity (or jet Reynolds number), producing rougher jets that trapped gas bubbles as they plunged through the liquid pool surface. Similarly, increasing the length of the free jet above the receiving pool increased the surface roughness and, therefore, increased $Q_a$. Increasing the nozzle diameter increased the diameter of the jet at the plunging jet point, and, therefore, gas was entrained over a longer perimeter, again increasing $Q_a$. The results shown here illustrate that $Q_a$ did not have a simple monotonic dependence on the downcomer diameter. For small downcomer diameters, $Q_a$ increased to a maximum with increasing $D_c$; as $D_c$ continued to increase, the liquid superficial velocity in the downcomer became so low that larger bubbles could rise upwards; the rising bubbles could then disentrain into the downcomer headspace. Thus, the *net* entrainment rate decreased as $D_c$ was increased further. The measured net air entrainment rate is increased due to the fact that the mesh hindered the coalescence phenomena and, hence, reduced the terminal velocity of the bubble, which increased the disentrainment rate. The air entrainment rate ratio, $Q_a/Q_j$, increased linearly with the nozzle contraction angle at high jet velocities; however, the linearity of the fit became poorer as the jet velocity decreased, as indicated by lower values of the correlation coefficient, $r^2$.

**Supplementary Materials:** The following are available online at http://www.mdpi.com/2073-4441/12/3/764/s1, Figure S1: Image shows very fine bubbles leaving the downcomer due to mesh effect.

**Funding:** This research was funded by the Kuwait Foundation for the Advancement of Sciences (KFAS).

**Acknowledgments:** The authors would like to thank the Kuwait Foundation for the Advancement of Sciences (KFAS) for their financial support through project No. P31475EC01.

**Conflicts of Interest:** The authors declare no conflict of interest.

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
