# Peer review of "Effect of Primary Variables on A Confined Plunging Liquid Jet Reactor"

_water, doi:10.3390/w12030764_

Round 1
Reviewer 1 Report
Dear Authors
We believe that your paper is very interesting. Overall the study was very good; there was nothing major that I felt needed comment. However, I would suggest some minor corrections, which are highlighted in the annotated version, I will send to you.
Best Regards

Reviewer 2 Report
Please find attached my comments.

Reviewer 3 Report
The authors studied a local vertical confined plunging liquid jet reactor (CPLJR) and investigated effects of operating conditions including a novel downcomer geometry on the gas/air entrainment rate. However, for the quality and novelty in the current paper, I do not think that this paper should be accepted in Water.
More discussion about the reactors used in wastewater treatment should be added in the paper to illustrate the difference in all the reactors. The full term of PLJR should be given in the main text. For Figure 1, the author cited "as shown in Figure 1(a) [2]." in the text, however, in the figure title, "Figure 1. (a) An unconfined plunging jet system (b) the same jet surrounded by a confining downcomer column [3]." Two different references??? The Introduction section should be rewritten and reconstructed. More recent papers should be discussed in the section. There is no Figure 2a, however, there is Figure 2b. The author did not prepare the paper very well. Figure 3 needs improvements. In Figure 5b, what are these symbols in the figure? Could you please provide four data points in Figure 6? The R2 is not very high. How many times of each experiment have you conducted at the same conditions? Please improve the quality of all the figures. What did you present in Figure 11? It is useless to this topic. it is hard to see the scale.Author Response
Please see the attachment

Round 2
Reviewer 2 Report
I think that the manuscript has not been revised properly. I believe that the main contributions of this work are: 1) presenting some experimental data for air entrainment problem; 2) demonstrating the effects of several geometrical parameters. For the first one, the authors simply give some descriptions of the data without an in-depth analysis of physics. For the second one, the authors emphasized many times the disentrainment, and so what? They had not provide any explanations. Some conclusions are obvious, any improvement? This is not sufficient to support the main purpose as noted in the manuscript title. Based on the above concerns, I do not recommend the publication of this manuscript in Water.
Author Response
Below are my responses to your comments.
I think that the manuscript has not been revised properly. I believe that the main contributions of this work are: 1) presenting some experimental data for air entrainment problem; 2) demonstrating the effects of several geometrical parameters. For the first one, the authors simply give some descriptions of the data without an in-depth analysis of physics. For the second one, the authors emphasized many times the disentrainment, and so what? They had not provide any explanations. Some conclusions are obvious, any improvement? This is not sufficient to support the main purpose as noted in the manuscript title. Based on the above concerns, I do not recommend the publication of this manuscript in Water.
- I think that the manuscript has not been revised properly
I have addressed all of your comments in round one (see round one comments).
- presenting some experimental data for air entrainment problem; 2) demonstrating the effects of several geometrical parameters
In the current study, I am not presenting an air entrainment problem, I am investigating the effect operating conditions on a CPLJR performance including a novel downcomer geometry. This will help me improving a CPLJR as an aerator.
- For the first one, the authors simply give some descriptions of the data without an in-depth analysis of physics.
I have provided an in-depth analysis of physics throughout the manuscript such as;
- Low velocity jets cause a depression of the free surface at the plunging points, entraining bubbles from the jet periphery via mechanisms. However, the radial inflow to the jet periphery is affected by the proximity of the confining walls.
- A vortex formed around the plunging point, which affected the shape of the surface depression around the jet.
- At higher Vj values, the surface of the jet became rough and pockets of gas were occluded as the liquid jet passed into the receiving pool.
- Generally, the dimensionless air entrainment rate ratio Qa/Qj increased with the nozzle contraction angle because of the increase in the surface roughness.
- The air entrainment rate ratio, Qa/Qj, increased linearly with the nozzle contraction angle at high jet velocities; however, the linearity of the fit became poorer as the jet velocity decreased, as indicated by lower values of the correlation coefficient, r2.
- At low jet velocities and for larger downcomer diameters, the superficial liquid velocity in the downcomer was insufficient to carry all of the entrained bubbles downwards, and thus the net entrainment rate decreased: a significant fraction of the bubbles was entrained and then rose up in the recirculation eddy and disentrained at the free surface.
- In the regions where the gas entrainment rate decreased as the downcomer diameter increased, the liquid superficial velocity was less than approximately 0.20-0.25 m/s, which corresponded to a typical bubble terminal velocity.
The foregoing points are examples of the in-depth analysis of physics that I provided in the current work.
- For the second one, the authors emphasized many times the disentrainment,
I will try to explain the point one more time. I am presenting CPLJR as an aerator that has several advantages over conventional aerators (e.g. mechanical aerators and diffusors). In this work I am interested in improving the performance of such system by increasing the net/measured entrainment rate (ejecting at the bottom of the downcomer) through reducing the disentrainment rate (unmeasured). In order to achieve this, I had to investigate the effect of the operating conditions to choose the best combinations of the operating conditions that give optimum results. Then I applied such conditions to the newly introduced downcomer (with mesh).
- and so what?
This comment isn’t relevant to the current work or science in general and hence I have no response to it.
Reviewer 3 Report
- Fig. 11 is still in a low quality. I would suggest the authors to move it to the SI.
- Please add more new references in this paper. Ref. [3] should be deleted from the paper.
Author Response
I would like to thank you for your 2nd round valuable comments and suggestions. Blow are my responses to your comments.
- Fig. 11 is still in a low quality.
Figure 11 is removed from lines 280-281 as shown by the “track change”. It is moved to SI as suggested by the reviewer. I have also added more information about how the mesh increased the net entrainment rate (lines – 278 – 279).
- Please add more new references in this paper. Ref.
The following references have been added, as suggested by the reviewer.
- Al-Anzi, B sh., Application of Confined Plunging Liquid Jet Reactor (CPLJR) as an aeration and brine dispenser technique for the environmental safe discharge of brine from Kuwait’s desalination plants, 2nd international conference on power and energy engineering, 2017, Munich, Germany, 63. (lines 330-332).
Cited in lines 33-35.
- Ajay Mandal, 2010, CHARACTERIZATION OF GAS-LIQUID PARAMETERS IN A DOWN-FLOW JET LOOP BUBBLE COLUMN, Brazilian Journal of Chemical Engineering Vol. 27, No. 02, pp. 253 – 264. (lines 369-371).
Cited in lines 111 – 112.
- Cummings, P.D., Chanson, H., 1997a. Air entrainment in the developing flow region of plunging jets – part 1: theoretical development. J. Fluids Eng. 119, 597–602. (lines 341-342).
Cited in lines 56 – 57.
- Miwa, S., Moribe, T., Tsutstumi, K., Hibiki, T., 2018. Experimental investigation of air entrainment by vertical plunging liquid jet. Chemical Engineering Science, 181: 251-263. (lines 346-347).
Cited in lines 56 – 57.
- [3] should be deleted from the paper.
Reference [3] has been removed (lines 328-329).
Round 3
Reviewer 2 Report
Overall the paper is enough